# Emerging Challenges in Methicillin Resistance of Coagulase-Negative Staphylococci

**DOI:** 10.3390/antibiotics14010037

**Published:** 2025-01-06

**Authors:** Marta Katkowska, Maja Kosecka-Strojek, Mariola Wolska-Gębarzewska, Ewa Kwapisz, Maria Wierzbowska, Jacek Międzobrodzki, Katarzyna Garbacz

**Affiliations:** 1Department of Oral Microbiology, Medical Faculty, Medical University of Gdansk, 80-204 Gdansk, Poland; marta.katkowska@gumed.edu.pl (M.K.); ewa.kwapisz@gumed.edu.pl (E.K.); maria.wierzbowska@gumed.edu.pl (M.W.); 2Department of Microbiology, Faculty of Biochemistry, Biophysics and Biotechnology, Jagiellonian University, 31-007 Krakow, Poland; maja.kosecka-strojek@uj.edu.pl (M.K.-S.); mariola.wolska@doctoral.uj.edu.pl (M.W.-G.); jacek.miedzobrodzki@uj.edu.pl (J.M.); 3Doctoral School of Exact and Natural Sciences, Jagiellonian University, 31-007 Krakow, Poland

**Keywords:** coagulase-negative staphylococci, CoNS, methicillin-resistance, borderline oxacillin resistance, *mec*A, cefoxitin, oxacillin, methicillin

## Abstract

**Objective:** In the present study, we used phenotypic and molecular methods to determine susceptibility to oxacillin in coagulase-negative staphylococci (CoNS) and estimate the prevalence of strains with low-level resistance to oxacillin, *mec*A-positive oxacillin-susceptible methicillin-resistant (OS-MRCoNS), and borderline oxacillin-resistant (BORCoNS). **Methods:** One hundred one CoNS strains were screened for oxacillin and cefoxitin susceptibility using phenotypic (disk diffusion, agar dilution, latex agglutination, and chromagar) and molecular (detection of *mec*A, *mec*B, and *mec*C) methods. Staphylococcal cassette chromosome *mec* (SCC*mec*) typing was performed. **Results**: Sixteen (15.8%) CoNS strains were *mec*A-positive, and 85 (84.2%) were *mec*-negative. Seven (6.9%) were classified as OS-MRCoNS, accounting for 43.8% of all *mec*A-positive strains. Twelve (11.9%) *mec*-negative strains were classified as borderline oxacillin resistant (BORCoNS). Compared with MRCoNS and BORCoNS, OS-MRCoNS strains demonstrated lower resistance to non-beta-lactams. SCC*mec* type I cassette was predominant. The disc-diffusion method with oxacillin accurately predicted OS-MRCoNS strains but did not provide reliable results for BORCoNS strains. Meanwhile, the latex agglutination test and CHROMagar culture accurately identified BORCoNS but not OS-MRCoNS. **Conclusions:** Finally, our findings imply that the recognition of methicillin resistance in CoNS requires a meticulous approach and that further research is needed to develop unified laboratory diagnostic algorithms to prevent the misreporting of borderline CoNS.

## 1. Introduction

Coagulase-negative staphylococci (CoNS) are a heterogeneous group comprising over 80 species. In contrast to coagulase-positive staphylococci, such as *Staphylococcus aureus*, CoNS have been considered less pathogenic or non-pathogenic for years. Most CoNS species colonize human skin and mucosal membranes and are rarely involved in clinically symptomatic infections [1,2]. However, given new groups of immunocompromised patients and the growing complexity of medical procedures, CoNS have emerged as a leading group of nosocomial pathogens nowadays [3]. CoNS often causes foreign body-associated infections and infections in preterm neonates [1,4,5]. While the infections are usually subacute and associated with only mild inflammation, they constitute a substantial clinical burden due to their widespread occurrence and problems with choosing an effective antibiotic therapy. Acquired resistance of CNoS to antimicrobial agents is a common problem and spreads quickly between species [5].

Among multiple mechanisms of acquired drug resistance displayed by CoNS, the resistance to beta-lactam antibiotics appears the most important from a clinical and epidemiological perspective [6]. The two principal mechanisms of staphylococcal resistance to beta-lactams include the production of enzymes, beta-lactamases, and mutated penicillin-binding proteins (PBP) [7]. Due to the generation of new PBP, with low affinity to beta-lactams, staphylococci become resistant to all beta-lactam antibiotics used in clinical practice nowadays, except for fifth-generation cephalosporins (ceftaroline and ceftobiprole) [8]. This type of resistance is referred to as methicillin or oxacillin resistance, as it was first documented in the case of methicillin, the first semisynthetic anti-staphylococcal penicillin [9]. CoNS displaying this type of resistance are called methicillin-resistant coagulase-negative staphylococci (MRCoNS). Methicillin resistance is determined by the presence of acquired *mec*A, *mec*B, or *mec*C genes (previously known as mecALGA251) that encode additional penicillin-binding proteins, PBP2a (PBP2’), PBP2b, or PBP2c, respectively [10,11]. *mec*A and *mec*C are chromosomal genes that are part of the staphylococcal cassette chromosome *mec* (SCC*mec*) [12].

Traditional phenotype-based methods have reduced sensitivity and specificity for the recognition of methicillin resistance in CoNS. In some cases, identifying the resistance can be challenging and inconclusive. For example, emerging oxacillin-susceptible methicillin-resistant *S. aureus* (OS-MRSA) strains were reported to carry *mec*A or *mec*C genes despite oxacillin MICs corresponding to sensitivity (up to 2 mg/L). Furthermore, some borderline oxacillin-resistant *S. aureus* (BORSA) strains may present with borderline oxacillin MICs yet not harbor the *mec* genes. Usually, the BORSA strains display low oxacillin (2–12 mg/L) and cefoxitin MICs (4–8 mg/L). On top of that, some authors use the term BORSA to describe both groups (OS-MRSA and BORSA) mentioned above, which makes their appropriate identification challenging and associated with a high risk of bias [13,14]. Meanwhile, accurate identification of methicillin (oxacillin) resistance is crucial for effective epidemiological intervention (eradication of MRSA) and implementation of a successful anti-staphylococcal therapy.

In principle, neither EUCAST nor CLSI guidelines recommend systematic screening of *Staphylococcus* spp. strains for borderline resistance to methicillin [15,16]. However, EUCAST recommends testing for the BORSA phenotype if methicillin resistance screening with cefoxitin disc (30 μg) yields a negative result, but oxacillin MIC exceeds 4 mg/L whenever clinical indications exist [15].

In the present study, we used phenotypic and molecular methods to determine susceptibility to oxacillin in various coagulase-negative staphylococci (CoNS) species and the occurrence of strains with low-level resistance to oxacillin, *mec*A-positive oxacillin-susceptible methicillin-resistant (OS-MRCoNS), and borderline oxacillin-resistant (BORCoNS) strains.

## 2. Results

Of the 101 coagulase-negative staphylococci, 16 (15.8%) strains were *mec*A-positive, and 85 (84.2%) did not carry any *mec* gene (*mec*A, *mec*B, or *mec*C) (Figure 1).

A comparative analysis of different methicillin resistance tests showed that oxacillin-based methods were highly sensitive in detecting *mec*A-positive phenotypes. Oxacillin disk diffusion and serial dilution methods showed 100% and 81.3% sensitivity, respectively, with PCR for *mec*A as a reference. The cefoxitin disk-diffusion test provided lower sensitivity (68.8%) but higher specificity (94.1%) than the oxacillin-based methods. Similarly, CHROMagar culture yielded lower sensitivity and higher specificity, 61.5% and 100%, respectively. The disc-diffusion method with oxacillin accurately predicted OS-MRCoNS strains but did not provide reliable results for BORCoNS strains. Meanwhile, the latex agglutination test and CHROMagar culture accurately identified BORCoNS but not OS-MRCoNS (Table 1).

### 2.1. Oxacillin-Susceptible Methicillin-Resistant Coagulase-Negative Staphylococci (OS-MRCoNS)

Seven (6.9%) *mec*A-positive strains demonstrated oxacillin MICs ≤ 0.5 mg/L or sensitivity to cefoxitin and/or oxacillin in the disk diffusion method. These strains, classified as oxacillin-susceptible strains (OS-MRCoNS), represented 43.8% of all *mec*A-positive strains. The OS-MRCoNS included *S. saprophyticus* (*n* = 3), *S. epidermidis* (*n* = 1), *S. haemolyticus* (*n* = 1), *S. pasteurii* (*n* = 1), and *S. warneri* (*n* = 1) species. All *S. saprophyticus* strains were resistant to oxacillin in the disk diffusion method but sensitive to cefoxitin. Other discrepancies in the results obtained with phenotypic methods are summarized in Table 2. Three OS-MRCoNS strains (*S. epidermidis*, *S. pasteurii*, and *S. saprophyticus*) carried type I SCC*mec*, one strain harbored type II SCC*mec*, and one strain presented with a unique type of SCC*mec* cassette. Three strains yielded false-negative results in the latex agglutination test, and four strains failed to grow on a CHROMagar (Table 2, Figure 1).

### 2.2. Methicillin-Resistant Coagulase-Negative Staphylococci (MRCoNS)

The remaining nine *mec*A-positive strains with oxacillin MICs of 1–24 µg/mL were classified as MRCoNS strains. All but one strain from this group produced PBP2a protein and demonstrated resistance to cefoxitin and oxacillin in the disc-diffusion method. *S. haemolyticus* (*n* = 5) was the most common species among the MRCoNS strains, followed by *S. hominis*, *S. epidermidis*, *S. saprophyticus*, and *S. warneri* (*n* = 1 each). The MRCoNS strains harbored predominantly type V (*n* = 3) and type I SCC*mec* (*n* = 1), with type II SCC*mec* carried by only one strain. All MRCoNS were correctly identified by latex agglutination test, and all but one were grown on a CHROMagar (Table 3, Figure 1).

### 2.3. Borderline Oxacillin-Resistant Coagulase-Negative Staphylococci (BORCoNS)

Twelve (11.9%) *mec*-negative strains with oxacillin MICs between 0.5 µg/mL and 2 µg/mL were classified as borderline oxacillin-resistant (BORCoNS). These strains accounted for 14.1% of all *mec*A-negative strains. Although none of the strains carried the *mec*A, *mec*B or *mec*C genes or produced PBP2a protein, almost all of them (91.6%) were resistant to oxacillin, and 41.6% displayed resistance to cefoxitin in the disc-diffusion method. The BORCoNS strains included *S. warneri* (*n* = 4), *S. equorum* (*n* = 2), *S. saprophyticus* (*n* = 2), *S. succinus* (*n* = 2), *S. cohnii* (*n* = 1), and *S. xylosus* (*n* = 1) species. *S. warneri* was the most frequently detected species among the BORCoNS identified, with all strains having oxacillin MICs of 0.5 µg/mL. Other discrepancies in the results obtained using phenotypic methods are summarized in Table 3. Moreover, the strains neither yielded positive results in the latex agglutination test nor grew on a CHROMagar (Table 4, Figure 1).

### 2.4. Antimicrobial Susceptibility Tests

All BORCoNS and MRCoNS strains produced penicillinase. Resistance to aminoglycosides and macrolides predominated in all groups of strains, displayed by approximately 50% of the isolates. Compared with MRCoNS and BORCoNS, OS-MRCoNS strains presented lower resistance to non-beta-lactam antibiotics, such as chloramphenicol, ciprofloxacin, clindamycin, and gentamycin. BORCoNS strains were more often resistant to tetracyclines (41.7%). All tested CoNS were susceptible to vancomycin, daptomycin, and linezolid (Table 5).

## 3. Discussion

Although CoNS have fewer virulence factors than *S. aureus*, they still play a significant role in opportunistic nosocomial infections, primarily due to the frequent occurrence of strains that constitute a reservoir of antimicrobial resistance [3]. Therefore, continuous attempts are undertaken at various levels of healthcare to accurately evaluate and monitor drug resistance, especially resistance to oxacillin, in CoNS [17]. CoNS strains with borderline resistance to oxacillin (OS-MRCoNS and BORCoNS) constitute a challenge for several reasons. First, EUCAST and CLSI guidelines defining laboratory standards for interpreting bacterial drug resistance are incomplete and inconclusive regarding CoNS [15,16]. Second, the strains with borderline resistance to oxacillin are difficult to detect and, as such, may be misreported as beta-lactam-sensitive or -resistant strains, mainly based on screening with phenotypic methods. This may result in prescribing antibiotics that are ineffective and eventually lead to treatment failure [18,19].

According to the recent EUCAST and CLSI guidelines, the disc-diffusion method with cefoxitin (30 µg) is a routine assay to identify resistance to oxacillin. This method is also considered a better predictor of *mec*A presence in most *Staphylococcus* spp. than the disc-diffusion method with oxacillin (1 μg) [15,16]. EUCAST recommends using the cefoxitin disc-diffusion method to identify nearly all CoNS species, with the differentiation between *S*. *epidermidis* and non-*epidermidis* strains based on species-specific zone diameters. The only exception pertains to *S. pseudintermedius*, in which EUCAST recommends using the oxacillin disc-diffusion test [15]. Similarly, CLSI advocates using the disc-diffusion method with cefoxitin for the identification of most species except for *S. pseudintermedius* and *S. schleiferi*, in the case of which oxacillin disc-diffusion test is recommended [16]. Regarding the borderline oxacillin MIC, EUCAST defined it at >2 mg/L for *S. aureus* and *S. lugdunensis* and at >0.25 for *S. epidermidis*. Unfortunately, the EUCAST guidelines do not specify borderline MICs for other CoNS species [15]. Meanwhile, CLSI defined borderline oxacillin MIC at ≥4 mg/L for *S. aureus* and *S. lugdunensis* and at ≥1 for the other CoNS, and this is the latter value that was used in our present study [16].

Due to the lack of a unified definition and diagnostic algorithm, microbiological laboratories face the problem of identifying and reporting CoNS strains with borderline resistance to oxacillin [20]. CLSI does not address the issue of borderline strains in their guidelines, whereas the EUCAST guidelines contain solely recommendations for borderline S*. aureus* (BORSA). Neither CLSI nor EUCAST guidelines provide recommendations for detecting borderline CoNS strains and further therapeutic approaches [15,16]. The lack of respective guidelines has been reflected by a substantial variety of interpretations and reporting of borderline resistance to oxacillin by various laboratories. Similarly, published research papers differ considerably in terms of used nomenclature, reported rates of borderline resistance, and approaches to treating infections caused by borderline-resistant pathogens.

*mec*A-positive oxacillin-sensitive *S. aureus* (OS-MRSA) strains have been reported in various geographical regions, including Europe, the United States, Brazil, Africa, Iran, India, and China [21,22,23,24,25,26]. The isolation rates of OS-MRSA from humans are typically estimated at a few to several percent (2–14.9%). The nearly 7% isolation rate documented in our present study fits within this range, but it needs to be emphasized that our current knowledge of the prevalence of OS-MRCoNS is considerably limited. In our present study, OS-MRCoNS strains belonged to various species, with the most common being *S. saprophyticus*, considerably less frequently found among classic MRCoNS, i.e., *mec*A-positive oxacillin-resistant strains. All OS-MR strains of *S. saprophyticus* were shown to be sensitive to cefoxitin on the disc-diffusion method and, hence, would be misreported as methicillin-sensitive (MSCoNS) on routine laboratory testing. Additionally, these strains yielded false negative results on CHROMagar, another factor contributing to their potential misidentification. Thus, despite being considered the most accurate predictor of *mec*A-positive strains, including *S. saprophyticus* strains, the disc-diffusion method with cefoxitin did not produce reliable results in our present study. Meanwhile, the disc-diffusion method with oxacillin accurately predicted the antibiotic resistance profile in most borderline strains. Misreporting OS-MRCoNS as methicillin-sensitive in routine laboratory practice may lead to failure in antibiotic therapy. Despite published reports about the effectiveness of beta-lactam antibiotics in treating experimentally induced infections with strains being borderline resistant to oxacillin, the application of this therapeutic option raises many concerns [27,28]. The resistance of OS-MRSA to oxacillin is considered inducible. It is postulated that after being exposed to this antibiotic, some OS-MRSA may transform into highly resistant clones insensitive to beta-lactams. The results of a recent study published by Gostev et al. suggest that the fast transformation of OS-MRSA to MRSA results from the preexistence of a small bacterial subpopulation with high MICs rather than from the selection of new mutants [29]. Variable levels of staphylococcal resistance to oxacillin may result from an instability in the DNA fragment that determines the resistance expression. The presence of a bacterial subpopulation with high MICs poses a high risk of transforming OS-MRSA into MRSA with a high level of resistance to beta-lactam antibiotics; this puts into question the usefulness of beta-lactams in the treatment of OS-MRSA infections. Thus, according to most authors, infections caused by OS-MRSA should be treated with antibiotics that are effective against MRSA, such as linezolid or vancomycin, rather than beta-lactams [27,29]. While, to the best of our knowledge, none of the published studies analyzed the problem in question with regards to OS-MRCoNS species, the same therapeutic approach seems applicable, especially given that compared to classic MRCoNS, the OS-MRCoNS isolated in our present study were more sensitive to non-beta-lactam antibiotics.

The second most frequently isolated group of strains with borderline resistance to oxacillin in the present study were BORCoNS (12%), i.e., *mec*A-negative strains showing resistance to oxacillin on phenotypic testing. According to the literature, the isolation rates of *S. aureus* with borderline resistance to oxacillin (BORSA) in a hospital setting vary from 1% to 12.5% or are even higher [19,30,31,32]. According to Khorvash et al., *mec*-negative oxacillin-resistant strains constituted up to 25.5% of all MRSA [33]. In another study, the isolation rate of BORSA from clinical material reached up to 50% [34]. The discrepancies in the isolation rates probably reflect the lack of unified diagnostic criteria for BORSA. The problem is even more evident in the case of BORCoNS strains that may be misidentified as methicillin-resistant and, thus, as non-eligible for beta-lactam antibiotic therapy. On the one hand, high-dose beta-lactams are considered an effective therapeutic option in non-complicated BORSA infections. On the other hand, Konstantinovski et al. demonstrated that such a therapy may be ineffective in severe infections, such as endocarditis, whereby beta-lactams should be substituted by vancomycin [35]. Similarly, Skinner et al. reported on a *mec*A-negative *S. aureus* strain with oxacillin MIC of 12 µg/mL, isolated from a patient with endocarditis, that did not respond to high-dose oxacillin therapy yet was successfully eradicated with vancomycin [36]. The BORCoNS isolated in our present study showed similar sensitivity to non-beta-lactam antibiotics, including vancomycin, as MRCoNS. Our findings imply that managing infections caused by *mec*-negative strains with borderline resistance to oxacillin requires a meticulous approach. The treatment protocol should be based on the results of antibiotic resistance testing, including oxacillin MIC.

Many SCC*mec* cassettes (I-VII) in coagulase-negative staphylococci were identified and reported. However, new combinations of the *ccr* and *mec* genes are detected; thus, unambiguous and appropriate identification is challenging. Additionally, it is indicated that those seven types described so far are not all types distributed worldwide. In our study, the SCC*mec* type I cassette was predominant. The occurrence of type I in many CoNS species has been described previously. The SCC*mec* type V was only found in *S. haemolyticus.* According to the literature, *S. haemolyticus* is a reservoir of SCC*mec* type V cassettes [37]. In our study, there were six strains with unknown SCC*mec* types, both OS-MRCoNS and MRCoNS. Many SCC*mec* cassettes found in CoNS cannot be classified as existing types, as they most likely contain undescribed allotypes or a mixture of existing ones [12].

According to EUCAST and CLSI guidelines, detection of the *mec* gene by PCR is considered a gold standard in evaluating methicillin resistance. It should be applied whenever the results of phenotypic testing are inconclusive, both for *S. aureus* and other *Staphylococcus* species. In routine screening, the disc-diffusion method with cefoxitin is considered the most accurate predictor of *mec*-positive staphylococci [15,16]. However, this method did not yield the desired predictive values in the present study, providing merely 68.8% sensitivity versus 100% sensitivity for the disc-diffusion method with oxacillin. Secchi previously reported the equally high sensitivity of the disc-diffusion method with oxacillin, whereas according to Swenson, this method produced a 94% sensitivity yet lower specificity (79%) [38,39]. Other authors also postulated using diagnostic criteria other than the disc-diffusion method with cefoxitin to minimize the risk of overlooking methicillin-resistant strains. According to Pinheiro et al., using multiple diagnostic criteria is remarkably advisable in the case of CoNS, which are more difficult to detect because of their heterogeneity and borderline resistance to oxacillin [17]. In our study, CHROMagar culture was the least sensitive method to detect *mec*-positive strains (61.5%). Other authors reported higher sensitivity to this test, but their studies involved *S. aureus* rather than CoNS and used CHROMagar from other manufacturers [40]. Notably, phenotypic methods, such as CHROMagar and latex agglutination method, accurately identified BORCoNS and MRCoNS as methicillin-sensitive and -resistant, respectively, providing 100% specificity. Identifying the resistance profile in OS-MRCoNS was more challenging, as the latex agglutination test and CHROMagar culture yielded false negative results. According to Nair, the sensitivity of the latex agglutination test was high (98.9%), with accurate positive results obtained for all OS-MRSA isolates [40]. The results of our present study and findings reported by other authors point to discrepancies between various tests and problems in the identification of borderline strains, implying that further research is needed before any harmonized recommendations could be published on this subject matter.

In summary, the present study demonstrated that CoNS strains with low levels of resistance to oxacillin may constitute a few to several percent of CoNS isolates and belong to various species. Our findings imply that the recognition of methicillin resistance in these strains requires a meticulous approach. The disc-diffusion method with oxacillin accurately predicted OS-MRCoNS strains but did not provide reliable results for BORCoNS strains. Meanwhile, the latex agglutination test and CHROMagar culture accurately identified BORCoNS but not OS-MRCoNS. Thus, the diagnostic protocol should be based not only on the results of phenotypic methods. The results presented herein warrant further research on borderline CoNS to develop unified laboratory diagnostic algorithms and to prevent the misidentification of these strains and antibiotic therapy failure.

## 4. Materials and Methods

### 4.1. Bacterial Strains

A total of 101 non-duplicate coagulase-negative staphylococci strains originating from the bacterial collection of the Department of Oral Microbiology, Medical University of Gdansk (MUG), were analyzed. The strains were isolated between 2016 and 2017, mainly from oral specimens during routine clinical laboratory procedures, not specifically for the present study. The collection included twelve CoNS species: *Staphylococcus warneri* (*n* = 43), *S. haemolyticus* (*n* = 12), *S. saprophyticus* (*n* = 9), *S. epidermidis* (*n* = 9), *S. pasteurii* (*n* = 8), *S. hominis* (*n* = 5), *S. xylosus* (*n* = 6), *S. equorum* (*n* = 3), *S. kloosii* (*n* = 2), *S. succinus* (*n* = 2), *S. cohnii* (*n* = 1), and *S. simulans* (*n* = 1). All strains were preliminarily identified using conventional methods. Identification of the strain at a species level was performed using the API system (bioMeriux, Marcy-l’Etoile, France) and confirmed by the matrix-assisted laser desorption ionization-time of flight mass spectrometry (MALDI-TOF) method (Bruker Daltonics, Bremen, Germany). The reference strains *S. aureus* ATCC 43300 and *S. aureus* ATCC 29213 were used as positive and negative controls, respectively. The isolates were stored at −80 °C in tryptic soy broth (Becton Dickinson, Franklin Lakes, NJ, USA) containing 20% glycerol.

### 4.2. Phenotypic Method of Methicillin Resistance Detection

#### 4.2.1. Disk Diffusion Method (DDM)

Methicillin resistance was identified using cefoxitin (30 μg) and oxacillin (1 μg) by the disk diffusion method on Mueller–Hinton agar (Becton Dickinson, Franklin Lakes, NJ, USA) per CLSI recommendations [16]. MHA plates were incubated at 35 to 37 °C at 16 to 18 h for oxacillin and 24 h for cefoxitin. Both the oxacillin and cefoxitin zones of inhibition were read using reflected light.

#### 4.2.2. Agar Dilution Method (ADM)

Oxacillin MICs were determined by agar dilution method according to CLSI recommendations [M07-A10]. All strains were plated on Mueller–Hinton agar supplemented with 2% sodium chloride. Accordingly, oxacillin was added to the media in concentrations in double dilution from 64 mg/L to 0.125 mg/L.

#### 4.2.3. PBP2a Latex Agglutination Test

To detect PBP2a expression, OXOID PBP2’ Latex Agglutination Test Kit (Basingstoke, UK) was used according to the manufacturer’s instructions. For *mecA*-positive strains with latex agglutination negative, the assay was repeated as recommended by the manufacturer following overnight oxacillin induction.

#### 4.2.4. CHROMagar MR

CHROMagar MR (GrasoBiotech, Starogard Gd., Poland) was used to screen methicillin-resistant strains. A 0.5 McFarland suspension of the bacterial colony prepared from a 24 h culture on 5% sheep blood agar was streaked on the CHROMagar MRSA. After 24 h of aerobic incubation at 35 °C, the grown blue colonies were indicated as MR strains.

### 4.3. Molecular Analysis of Methicillin Resistance

#### 4.3.1. Genomic DNA Extraction

For genomic DNA extraction, strains were grown for 20 h at 37 °C on blood agar plates. A full inoculation loop of 10 μL of bacterial colonies was homogenized with a TissueLyser II (Qiagen, Germantown, MD, USA). The Qiagen DNeasy Blood & Tissue Kit (Qiagen, Germantown, MD, USA) was used for genomic DNA extraction. The subsequent steps were performed according to the manufacturer’s instructions. Purified DNA was stored at −20 °C.

#### 4.3.2. Detection of *mec*A, *mec*B, and *mec*C Genes

To detect the *mec*A gene, primers *mec*A-f1: TGGCCAATACAGGAACAGCA; *mec*A-r1: ACGTTGTAACCACCCCAAGA were designed for *S. cohnii*, *S. epidermidis*, *S. equorum*, *S. haemolyticus*, *S. hominis*, *S. saprophyticus*, *S. succinus*, *S. warneri*, and *S. xylosus* species (based on selected species with following GenBank accession numbers: NZ_CP073878.1; NZ_CP073863.1; CP045187.2; NZ_CP035541.1; NZ_CP065797.1; NZ_CP093539.1; CP014567.1). For *mec*A-negative strains, *mec*C and *mec*B genes were tested by simplex PCR according to Ito et al. and Becker et al., respectively [41,42].

#### 4.3.3. Detection of SCC*mec* Cassettes

The SCC*mec* cassettes were typed using two independent methods described by Milheirico et al. [43] and Kondo et al. [44], with the USA300 3956/13 strain as a positive control. The PCR products were resolved by electrophoresis, and band patterns were analyzed.

#### 4.3.4. Antimicrobial Susceptibility Testing

The susceptibility of CoNS isolates to antimicrobial agents was determined by the disk diffusion method, according to the European Committee on Antimicrobial Susceptibility Testing (EUCAST). The following antimicrobial agents were used for the test: amoxicillin/clavulanic acid, chloramphenicol, ciprofloxacin, clindamycin, erythromycin, gentamicin, penicillin, tetracycline, trimethoprim/sulfamethoxazole (Oxoid, Basingstoke, UK). The susceptibility to vancomycin, daptomycin, and linezolid was determined by E-test. The multidrug-resistant (MDR) strains showed resistance to at least three classes of antibiotics. The inducible macrolide–lincosamide–streptogramin B (MLS_B_) resistance was detected by the D-test and interpreter according to the EUCAST. The penicillinase encoded by the blaZ gene was detected by the cefinase discs test (Becton, Dickinson and Company, Drogheda, Ireland).

## Figures and Tables

**Figure 1 antibiotics-14-00037-f001:**
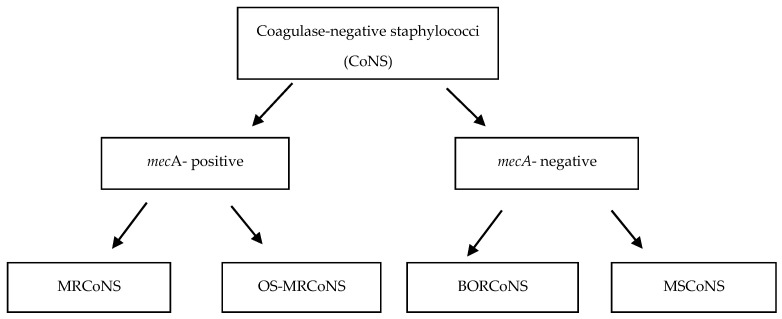
Detection of methicillin resistance in coagulase-negative staphylococci (CoNS).

**Table 1 antibiotics-14-00037-t001:** Screening of methicillin resistance in coagulase-negative staphylococci (CoNS) strains.

Methods	Sensitivity (%)	Specificity (%)	PPV (%)	NPV (%)
OXA DDM	100	87.1	57.7	100.0
FOX DDM	68.8	94.1	68.8	94.1
OXA ADM	81.3	90.6	61.9	96.3
CHROMagar	61.5	100.0	100.0	94.4
PBP2A	81.3	100.0	100.0	96.6

FOX—cefoxitin; OXA—oxacillin; DDM—disk diffusion method; ADM—agar dilution method; PBP—penicillin-binding protein; PPV—positive predictive value; NPV—negative predictive value.

**Table 2 antibiotics-14-00037-t002:** Phenotypic and genotypic characteristics of *mec*A-positive oxacillin-susceptible methicillin-resistant coagulase-negative staphylococci (OS-MRCoNS).

Species	Disc Diffusion		OXA MIC [mg/L]	PBP2A	*mec*A	SCC*mec*	Antimicrobial Resistance Profile	**Chromagar MR**
	FOX	OXA						
*S. epidermidis*	S	R	2	+	+	I	OXA-ERY	+
*S. haemolyticus*	R	S	1	−	+	unique	FOX-ERY	+
*S. pasteurii*	R	R	0.5	+	+	I	FOX-OXA-ERY-GMN	−
*S. saprophyticus*	S	R	1	−	+	II	OXA-GMN	−
*S. saprophyticus*	S	R	0.5	+	+	NT	OXA-ERY-TET-SXT	−
*S. saprophyticus*	S	R	1	+	+	I	OXA	−
*S. warneri*	R	R	0.5	−	+	NT	FOX-OXA-GMN-P	+

FOX—cefoxitin, OXA—oxacillin, ERY—erythromycin, GMN—gentamycin, P—penicillin, SXT—trimethoprim/sulfamethoxazole, TET—tetracyclin, PBP2A—penicillin-binding protein, *mec*A—gene of methicillin resistance, SCC*mec*—staphylococcal cassette chromosome *mec*, NT—non-typeable.

**Table 3 antibiotics-14-00037-t003:** Phenotypic and genotypic characteristics of *mec*A-positive methicillin-resistant coagulase-negative staphylococci (MRCoNS).

Species	Disc Diffusion		OXA MIC [mg/L]	PBP2A	*mec*A	SCC*mec*	Antimicrobial Resistance Profile	**Chromagar MR**
	FOX	OXA						
*S. epidermidis*	R	R	32	+	+	IV	FOX-OXA-ERY-CMN-TET-GMN-SXT	+
*S. haemolyticus*	S	R	32	+	+	NT	OXA-ERY-CMN-GMN	+
*S. hominis*	R	R	1	+	+	I	FOX-OXA-ERY-P-SXT	+
*S. haemolyticus*	R	R	32	+	+	V	FOX-OXA	+
*S. haemolyticus*	R	R	32	+	+	V	FOX-OXA-ERY-TET-GMN	+
*S. haemolyticus*	R	R	32	+	+	I	FOX-OXA-ERY-CIP-GMN-SXT	+
*S. saprophyticus*	R	R	2	+	+	NT	FOX-OXA	−
*S. haemolyticus*	R	R	32	+	+	V	FOX-OXA-ERY-CIP-TET-GMN	+
*S. warneri*	R	R	1	+	+	unique	FOX-OXA-GMN	+

FOX—cefoxitin, OXA—oxacillin, ERY—erythromycin, GMN—gentamycin, P—penicillin, SXT—trimethoprim/sulfamethoxazole, TET—tetracyclin, PBP2A—penicillin-binding protein, *mec*A—gene of methicillin resistance, SCC*mec*—staphylococcal cassette chromosome *mec*, NT—non-typeable.

**Table 4 antibiotics-14-00037-t004:** Phenotypic and genotypic characteristics of borderline oxacillin-resistant coagulase-negative staphylococci strains (BORCoNS).

Species	Disc Diffusion		OXA MIC [mg/L]	PBP2A	*mec*A	Antimicrobial Resistance Profile	**Chromagar MR**
	FOX	OXA					
*S. cohnii*	S	R	2	-	-	OXA-ERY-P	-
*S. equorum*	S	R	1	-	-	OXA-ERY-CMN-P	-
*S. equorum*	R	R	1	-	-	FOX-OXA	-
*S. saprophyticus*	S	R	2	-	-	OXA-CHL-ERY-TET-P	-
*S. saprophyticus*	R	R	1	-	-	FOX-OXA-CHL-ERY-TET-P	-
*S. succinus*	S	R	1	-	-	FOX-OXA-ERY-CMN-GMN-P	-
*S. succinus*	R	R	1	-	-	FOX-OXA-GMN-TET-P	-
*S. warneri*	S	R	0.5	-	-	OXA	-
*S. warneri*	R	S	0.5	-	-	FOX-GMN-TET-P	-
*S. warneri*	S	R	0.5	-	-	OXA-ERY-GMN-P	-
*S. warneri*	R	R	0.5	-	-	FOX-OXA-GMN-TET-P	-
*S. xylosus*	S	R	1	-	-	OXA-GMN-P	-

FOX—cefoxitin, OXA—oxacillin, CMN—clindamycin, ERY—erythromycin, GMN—gentamycin, P—penicillin, TET—tetracyclin, PBP2A—penicillin-binding protein.

**Table 5 antibiotics-14-00037-t005:** Comparison of antimicrobial-resistance of OS-MRCoNS MRCoNS and BORCoNS strains.

Antimicrobial Agents	OS-MRCoNS (*n* = 7)	MRCNS (*n* = 9)	BORCoNS (*n* = 12)
chloramphenicol	0 (0.0%)	0 (0.0%)	2 (16.7%)
ciprofloxacin	0 (0.0%)	2 (22.2%)	0 (0.0%)
clindamycin	0 (0.0%)	2 (22.2%)	2 (16.7%)
erythromycin	4 (57.1%)	6 (66.7%)	6 (50.0%)
gentamycin	3 (42.9%)	6 (66.7%)	6 (50.0%)
penicillin	7 (100%)	8(88.9%)	12 (100%)
trimethoprim/sulfamethoxazole	1 (14.3%)	3 (33.3%)	0 (0.0%)
tetracyclin	1 (14.3%)	3 (33.3%)	5 (41.7%)
daptomycin	0 (0.0%)	0 (0.0%)	0 (0.0%)
linezolid	0 (0.0%)	0 (0.0%)	0 (0.0%)
vancomycin	0 (0.0%)	0 (0.0%)	0 (0.0%)

## Data Availability

Data are contained within the article.

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
