# Peer review of "Emerging Challenges in Methicillin Resistance of Coagulase-Negative Staphylococci"

_antibiotics, 2025, doi:10.3390/antibiotics14010037_

Round 1

Reviewer 1 Report

Comments and Suggestions for Authors

The authors describe a study with the stated aim of using a variety of established phenotypic and  genotypic methods (mec PCRs and pvl) to determine methicillin resistance in various species of coagulase-negative staphylococci (CoNS). Due to the lack of a gold standard the study was essentially a comparison of the outcomes of the various assays. In the absence of a gold standard conclusions regarding the accuracy of each test were confusing and lacked support. The samples were limited in that they only included 16 mecA PCR positives. Nonetheless methicillin resistant CoNS and the difficulty identifying them is an important area for which there is a dearth of information.

I have the following recommendations:

1.      Line 79: Remove “prevalence”. There are too few samples of any one species to estimate prevalence in their population and the term is not used elsewhere in the manuscript.

2.      Line 133: PCR detection of mecA is a key element of this study. The authors state that their mecA PCR primers were designed based on genomes from 10 species of staphylococci and list the respective GenBank Accession numbers for the species. However, the listed species do not correspond to the accession numbers and (based on my NCBI search) entries are only included for four separate species as follows:  NZ_CP073878.1; Staphylococcus epidermidis,  NZ_CP073863.1; Staphylococcus epidermidis, CP045187.2; Staphylococcus haemolyticus, NZ_CP035541.1; Staphylococcus haemolyticus, NZ_CP065797.1; Staphylococcus saprophyticus, NZ_CP093539.1;  Staphylococcus hominis, and  CP014567.1; Staphylococcus hominis. The actual species used for designing the primers and their correspondence with mecA from the species studied must be addressed.

3.      Line 144: Many species of staphylococci have cytotoxins distinct from S. aureus pvl and all samples were negative with the primers used to detect it.  The primer pair used in this study needs to be validated for the target species to avoid implying that they lack similar virulence proteins and the rationale for attempting to detect it should be included.

4.      Line 179: Table 1. The terms “specificity” and “sensitivity” are problematic without a gold standard. The authors should explain the basis for their determinations here and throughout the manuscript.

5.      Line 192: “unique” or “previously unreported” type of SCCmec would be preferable to “unusual”.

6.      Line 197: Table 2 is confusing as its title includes “oxacillin-susceptible methicillin resistant” and lists, based on disc diffusion, a sample that is resistant to both cefoxitin and oxacillin, and is positive on the chromagar MR test, but is classified as oxacillin susceptible based on OX MIC. Again, this seems to be an issue of defining methicillin resistance.

7.      Line 301: The discussion of methicillin resistance inducibility and transformation from the susceptible to resistant phenotype should include the analysis of Gostev et al [34] and others regarding mutations in the mecA promoter region, mutations in genes associated with metabolic pathways, and mecA accessory gene products (mecR1 and mecI).

Author Response

Response to Reviewer 1 Comments

Dear Reviewer,

First, we would like to thank you for a detailed review of our manuscript and all of your valuable remarks. We have addressed them all in detail below. We hope that you will find our explanations and manuscript modifications sufficient for reconsidering its publication in the Antibiotics.

With kind regards,

Authors

Comments 1: Line 79: Remove “prevalence”. There are too few samples of any one species to estimate prevalence in their population and the term is not used elsewhere in the manuscript.

Response 1: The sentence was changed in the revised manuscript, line 79.

Comments 2: Line 133: PCR detection of mecA is a key element of this study. The authors state that their mecA PCR primers were designed based on genomes from 10 species of staphylococci and list the respective GenBank Accession numbers for the species. However, the listed species do not correspond to the accession numbers and (based on my NCBI search) entries are only included for four separate species as follows:  NZ_CP073878.1; Staphylococcus epidermidis,  NZ_CP073863.1; Staphylococcus epidermidis, CP045187.2; Staphylococcus haemolyticus, NZ_CP035541.1; Staphylococcus haemolyticus, NZ_CP065797.1; Staphylococcus saprophyticus, NZ_CP093539.1;  Staphylococcus hominis, and  CP014567.1; Staphylococcus hominis. The actual species used for designing the primers and their correspondence with mecA from the species studied must be addressed.

Response 2: We are very grateful for this comment and agree with the Reviewer. The corrected paragraph has been updated accordingly in lines 133-138 of the revised manuscript. We would also like to clarify that the primers were specifically designed based on the genomes of four species (S. epidermidis, S. haemolyticus, S. hominis, and S. saprophyticus), as these species had annotated mecA genes in their genomes according to the GenBank database (two references per species). For the remaining species (S. warneri, S. equorum, S. cohnii, S. succinus, and S. xylosus), we were unable to find genomes of methicillin-resistant strains in GenBank. We verified this by checking annotations and relevant literature. Although there are publications reporting methicillin-resistant strains of these species with mecA and cassette types, these studies do not perform whole-genome sequencing of the strains. Consequently, no genomic references are available for these species.

Comments 3: Line 144: Many species of staphylococci have cytotoxins distinct from S. aureus pvl and all samples were negative with the primers used to detect it.  The primer pair used in this study needs to be validated for the target species to avoid implying that they lack similar virulence proteins and the rationale for attempting to detect it should be included.

Response 3: We appreciate this comment. In our study, the primers used to detect pvl were originally designed for S. aureus and have not been validated for coagulase-negative Staphylococcus species. We understand that this limitation may lead to ambiguity in interpreting the results. Given Rewiewer’s valuable comment and to avoid potential misinterpretation, we have decided to remove the section on pvl detection from all sections of the manuscript. Our initial rationale was to explore the potential horizontal gene transfer of S. aureus pvl to coagulase-negative staphylococci; validating primers for homologous virulence genes across other species was beyond the scope of this study. We believe removing this section will improve the clarity of the manuscript.

Comments 4: Line 179: Table 1. The terms “specificity” and “sensitivity” are problematic without a gold standard. The authors should explain the basis for their determinations here and throughout the manuscript

Response 4: According to EUCAST and CLSI guidelines, detecting the mec gene by PCR is considered the gold standard for assessing staphylococcal methicillin- resistance. We have assumed that the detection of the mec gene was the reference method to which we relate the results of phenotypic tests. It was necessary to compare the results of various techniques commonly used in laboratories to detect methicillin-resistance.

Comments 5: Line 192: “unique” or “previously unreported” type of SCCmec would be preferable to “unusual”.

Response 5: The word has been changed in the text, and in the Table 2 and Table 3.

Comments 6: Line 197: Table 2 is confusing as its title includes “oxacillin-susceptible methicillin resistant” and lists, based on disc diffusion, a sample that is resistant to both cefoxitin and oxacillin, and is positive on the chromagar MR test, but is classified as oxacillin susceptible based on OX MIC. Again, this seems to be an issue of defining methicillin resistance.

Response 6:  OS-MRSA/ OS-MRCoNS is an international abbreviation for oxacillin-susceptible methicillin resistant staphylococcal strains that were described in the cited literature (References: 25-30).

Comments 7: Line 301: The discussion of methicillin resistance inducibility and transformation from the susceptible to resistant phenotype should include the analysis of Gostev et al [34] and others regarding mutations in the mecA promoter region, mutations in genes associated with metabolic pathways, and mecA accessory gene products (mecR1 and mecI).

Response 7:

We greatly appreciate Reviewer comment. We agree that discussing the inducibility of methicillin resistance (MR) and the transformation from the susceptible to resistant phenotype could have included a more detailed analysis, however, we have dedicated a paragraph to this in the Discussion (lines 298-307), and our study was aimed at screening different methods for detecting MR in CoNS.

Reviewer 2 Report

Comments and Suggestions for Authors

 In this article Katkowska and coworkers characterized methicillin resistance in 101 coagulase-negative staphylococci (CoNS) species using molecular and phenotypic methods. Overall, the study is well conducted and the analysis well performed. I have the following comments for the authors:

1. Ensure that all bacterial and gene names in the material and methods section “Detection of mecA, mecB, and mecC gene” are italicized.

2. The antibiotic code for oxacillin should be revised from “OX” to the three-letter code “OXA” through the manuscript.

3. In Table 2, the antimicrobial resistance profile for S. warneri is “FOX-OX-GMN-P” what does the P represent?

4. Table 3: The abbreviated three letter code “CMN” is not described in the Table footnote

5. Table 4:  Again, the abbreviated three letter code “CMN” is not described in the Table footnote. What does “P” in the antimicrobial resistance profiles in Table 4 stand for?

6. Line 375 “in this matter” consider revising to “on this subject matter”

7. Reports in the literature suggest that infections caused by BORSA strains can be treated with beta-lactam antibiotics especially BORSA isolates that display low oxacillin MICs (≤2µg/ mL). In Table 4, all the oxacillin MIC values were ≤2 µg/ mL however, the susceptibility of these strains to beta-lactam antibiotics was not evaluated. Can the authors comment on why the susceptibility of these strains to beta-lactam antibiotics was not investigated?

Author Response

Response to Reviewer 2 Comments

Dear Reviewer,

First, we would like to thank you for a detailed review of our manuscript and all of your valuable remarks. We have addressed them all in detail below. We hope that you will find our explanations and manuscript modifications sufficient for reconsidering its publication in the Antibiotics.

With kind regards,

Authors

Comments 1: Ensure that all bacterial and gene names in the material and methods section “Detection of mecA, mecB, and mecC gene” are italicized.

Response 1: The changes were made in the section: “Detection of mecA, mecB, and mecC gene”, lines 132-141.

Comments 2: The antibiotic code for oxacillin should be revised from “OX” to the three-letter code “OXA” through the manuscript.

Resposne 2: Changes have been made throughout the manuscript.

Comments 3: In Table 2, the antimicrobial resistance profile for S. warneri is “FOX-OX-GMN-P” what does the P represent?

Comments 4: Table 3: The abbreviated three letter code “CMN” is not described in the Table footnote

Comments 5: Table 4:  Again, the abbreviated three letter code “CMN” is not described in the Table footnote. What does “P” in the antimicrobial resistance profiles in Table 4 stand for?

Resposne 3-5: Explanations of abbreviations have been added in the manuscript.

Comments 6: Line 375 “in this matter” consider revising to “on this subject matter”

Resposne 6: The change was reflected in the text, lines 377-388.

Comments 7: Reports in the literature suggest that infections caused by BORSA strains can be treated with beta-lactam antibiotics especially BORSA isolates that display low oxacillin MICs (≤2 µg/ mL). In Table 4, all the oxacillin MIC values were ≤2 µg/ mL however, the susceptibility of these strains to beta-lactam antibiotics was not evaluated. Can the authors comment on why the susceptibility of these strains to beta-lactam antibiotics was not investigated?

Response 7: We greatly appreciate Reviewer comment, penicillin-resistance has shown that BORSA produces β-lactamases.

Round 2

Reviewer 1 Report

Comments and Suggestions for Authors

The authors have addressed my concerns.